# Utility of Condition Indices as Predictors of Lipid Content in Slimy Sculpin (*Cottus cognatus*)

**Adrian R. Hards** \*, **Michelle A. Gray**, **Sophia C. Noël and Rick A. Cunjak**

Canadian Rivers Institute and the Department of Biology, University of New Brunswick, 10 Bailey Drive, Fredericton, NB E3B 5A3, Canada; mgray1@unb.ca (M.A.G.); clara.maria@unb.ca (S.C.N.); cunjak@unb.ca (R.A.C.)

\* Correspondence: adrian.hards@unb.ca; Tel.: +1-506-452-6204

**Abstract:** Slimy sculpin (*Cottus cognatus*) are increasingly being used as indicator species. This has primarily entailed measuring their condition, the assumption being that condition can be used as a surrogate for lipid content. While there is evidence to suggest this assumption is applicable to some fish, it has yet to be validated for *C. cognatus*. Further, there are several means by which one may calculate condition, the most commonly employed of which are indirect measurements of lipid content (namely, Fulton's *K*, somatic *K* (*Ks*), and Le Cren's relative condition factor (*Kn*)). We compared the ability of each of these morphometric indices to predict whole-body lipid content in *C. cognatus*. There was a moderate degree of evidence that Fulton's *K*, *Ks*, and *Kn* are reliable predictors (*Ks* and *Kn* in particular). Of the latter we recommend *Kn* be used because, unlike *Ks*, it does not require that fish be killed. And while Fulton's *K* did not perform quite as well, we consider it a sufficient substitute if the data necessary to calculate *Kn* are unavailable.

**Keywords:** *Cottus cognatus*; Fulton's condition factor; lipid content; morphometrics; relative condition factor; slimy sculpin

## 1. Introduction

Slimy sculpin *Cottus cognatus* (Richardson 1836) is a small-bodied fish that inhabit lakes and rivers throughout Canada, the northeastern United States and Alaska [1]. They are often found in high densities, have small home ranges [2,3] and are iteroparous total spawners [4–6]. These particular traits have been cited by researchers to justify their use of *C. cognatus* as an environmental indicator species [7] (for example, in the monitoring of pulp and paper mill discharge [8], agricultural runoff [9], oil sands operations [10] and hydropower generation [11,12]).

The use of *C. cognatus* in such instances typically entails measuring their condition, a widely used but often undefined term that generally refers to an individual's overall health [13]. Itself an ambiguous term, health is commonly used to refer to the amount of lipid present in an individual (specifically, a subgroup of lipids called triglycerides, a type of fat that is the principal energy storage substrate in fish) [14–17]. The accumulation of fat is an adaptation to living in seasonal environments. It is accumulated during the growing season and depleted when energy expenditure exceeds intake to prevent starvation-related mortality (for example, during the winter and gonadal development). Therefore, it is unsurprising that fat content has been found to be a strong predictor of fish survival and reproductive success, or in more general terms, health [18].

One of the most frequently used methods of measuring lipid content is that of Bligh and Dyer [19]. This involves total lipid extraction from whole fish. Such direct methods can, however, be prohibitively time consuming and expensive when applied to a large fish sample or when utilised in long-term

studies. They also require that fish be killed, which, in addition to being ethically undesirable, may limit the scope of future research in the areas under study.

Thankfully, there exist several indirect condition indices that avoid these shortcomings, the most common of which are based on two simple measurements: total length and wet mass. These are used to calculate what are often referred to as morphometric indices [20], two of the most commonly used of which are Fulton's condition factor (Fulton's *K*) [21], and Le Cren's [22] relative condition factor (*Kn*) (see "Materials and Methods"). Finally, there is the somatic *K* (*Ks*). This is calculated in the same way as Fulton's *K* (see "Materials and Methods") but uses somatic mass as opposed to total mass to eliminate the confounding effects of stomach contents and gonads.

These provide a measure of the degree of plumpness of an individual that is assumed to be positively related to its lipid reserves. However, the assumption that changes in fat reserves result in measurable changes in plumpness has yet to be validated for *C. cognatus*, and while there is some evidence to suggest this may be true for certain species (for example, Atlantic salmon *Salmo salar* (Linnaeus 1758) [23]), it would be careless to extrapolate to all fish [24]. Indeed, there exist examples where such morphometric indices were observed to be poorly correlated with whole-fish lipid content within the same family as the previous example (specifically, Chinook salmon *Oncorhynchus tshawytscha* (Walbaum 1792) [25]). If validated, such indices provide a rapid and inexpensive means by which to answer a variety of questions, such as how anthropogenic stressors (like sewage discharge and pulp mill effluent [8]), affect growth, and how condition can influence fecundity [26,27], population dynamics [28], and predation pressure [29].

The primary objective of our study was to assess the ability of three commonly used condition indices (namely: Fulton's *K*, *Ks*, and *Kn*) to predict whole-body lipid content in *C. cognatus* as estimated by the Bligh and Dyer [19] method.

Fulton's *K* assumes isometric growth (specifically, that body mass increases as the cube of total length [30,31]). Unfortunately, this assumption is rarely valid. Instead, fish often exhibit positive allometric growth, i.e., they become more rotund than the increase in length would imply if isometric growth were occurring [21,31,32]. The use of Fulton's *K* on a species that exhibits such growth would, therefore, result in overestimates of their condition. Fulton's *K* is also susceptible to variations in stomach content and gonadal mass which are independent of fat reserves [22]. This too may lead to inaccurate estimates of condition. Therefore, we expected the relationship between lipid content and *Ks* to be stronger than that between lipid content and Fulton's *K*.

While Fulton's *K* measures deviations from an individual that holds to the 'cube law', *Kn* (i.e., relative condition factor) measures deviations from the expected mass (predicted by a mass-length equation derived from historical data specific to the population or region being examined). This eliminates the assumption of isometric growth [20]. As a result, some studies have found *Kn* to be an improvement over Fulton's *K* as a measure of body condition [33]. Consequently, we expected the relationship between *Kn* and lipid content to be stronger than that for Fulton's *K*. However, *Kn* is still influenced by variations in stomach content and gonadal mass. We therefore predicted that among the indirect indices being compared, either *Kn* or *Ks* would be the best at predicting lipid content and that Fulton's *K* would be the worst predictor.

Finally, the distribution of energy and the role of lipids may change as *C. cognatus* grow. Juveniles, for instance, often grow in length at a greater ratio than they do in body mass [31]. This would affect the suitability of condition indices for estimating lipid content. As a result, we hypothesized that the maturity status of individuals would influence the utility of condition indices to predict lipid content.

## 2. Materials and Methods

Sampling was conducted in the Tobique River basin, New Brunswick, Canada. The study area consisted of four wadeable streams, two of which experience flow regulation (the River Dee (47°01′45.5″ N 66°58′49.9″ W and 47°07′20.5″ N 67°00′24.5″ W) and the Serpentine River (47°09′43.3″ N 66°51′53.8″ W and 47°12′52.2″ N 66°51′01.1″ W)) and two of which were unregulated

(the Wapske River (46°52′14.0″ N 67°06′58.4″ W and 46°52′11.7″ N 67°16′20.9″ W) and the Gulquac River (46°56′10.0″ N 67°04′01.6″ W and 46°58′46.3″ N 67°14′47.6″ W)). Two study reaches were selected along each stream, and in each reach three successive riffles were sampled. Single-pass backpack electrofishing (settings: 500 V, 50–60 Hz; Smith-Root LR-24, Vancouver, Washington) was conducted in an upstream direction without the use of block nets between the hours of 10:00 and 16:00. Riffles were subjectively identified as areas with swift flowing water (>0.3 m/s), shallow depth (<0.5 m), and a high proportion of broken water surface owing to the boulder and cobble dominated substrate (as defined by Wentworth [34]). The field crew consisted of one person with an electrofisher, a dip-netter, and two people holding a lightly-modified Pollett lip seine (3 m wide; 2 mm mesh) [35].

Owing to a scarcity of historical data for the region, *C. cognatus* were collected each October, 2013–2015, for the purpose of calculating *Kn*. A total of 1446 fish were measured to the nearest 1 mm and weighed to the nearest 0.01 g after removal of excess water with paper towels. All fish were released. In October 2015, additional *C. cognatus* were killed for lipid extraction, thereby creating two datasets. Hereinafter, we refer to these respective datasets as "non-lethal" and "lethal", respectively. *C. cognatus* caught for lipid extraction did not contribute to the non-lethal sample data.

*C. cognatus* that were captured for lipid extraction were divided into four total length categories, namely: 20 to 40 mm; 41 to 60 mm; 61 to 80 mm; and 81 to 100+ mm to ensure a variety of sizes were obtained. Up to three *C. cognatus* were haphazardly selected from each length category. Length and mass measurements were taken as above in situ to best replicate what we presume is most commonly practised by other researchers. *C. cognatus* were subsequently killed by an overdose of tricaine methanesulfonate (MS-222), a sharp blow to the occiput, and spinal severance. Once killed, fish were dissected to determine their sex and remove their stomach contents and gonads. After the stomach contents and gonads were excised, each carcass was weighed to determine its somatic mass. Any abnormalities (namely, lesions, tumours, or parasites) were recorded. Carcasses were placed in a portable −20 °C freezer for the remainder of the sampling period (at most three days). Upon returning from the field, all individuals were transferred to a −80 °C freezer and stored therein for 12 months before lipid extraction. A total of 96 *C. cognatus* were killed for lipid extraction (12 fish from each of the two study reaches on each of the four rivers), however three individuals were discarded owing to the presence of parasites. The resultant *n* was 68 for adults and 25 for juveniles.

This experimental protocol was reviewed by the University of New Brunswick (UNB) Animal Care Committee to ensure it adhered to Canadian Council of Animal Care (CCAC) guidelines (#15019).

*2.1. Condition Indices and Lipid Extraction*

From the aforementioned measurements, three condition indices were calculated for each fish: Fulton's *K*, *Ks*, and *Kn*. Fulton's *K* was calculated as per Ricker [21]:

$$K = (\frac{M}{L^3}) \times 10^N \tag{1}$$

where *M* is total wet mass (g), *L* is total length (mm), and *N* is an integer that brings K near 1 (which in our study was 5 owing to the units of measurement used). *Ks* was calculated in largely the same manner as Fulton's *K*, but by using somatic mass (*S*) (g) as opposed to total wet mass (*M*):

$$K_S = (\frac{S}{L^3}) \times 10^N \tag{2}$$

*Kn* was calculated as:

$$Kn = (\frac{M}{m}) \tag{3}$$

where *M* is the observed total wet mass (g) of an individual and *m* is the total wet mass that an individual of the same length is expected to have, as calculated from a mass-length regression equation

(see "Statistical Analyses" below). Note, there is also a somatic equivalent of *Kn,* but we were unable to calculate it because we did not conduct lethal sampling prior to 2015.

Whole fish carcasses, including stomach contents and gonads, were oven dried at 60 °C until a constant mass was achieved and then homogenized using a mortar and pestle. Sub-samples consisting of 200 ± 5 mg of fish homogenate were used for lipid extraction. Lipids were extracted using a water, chloroform and methanol solution (the final ratio of which was 1.8:2:2 as specified by Bligh and Dyer [19]). Phase separation was achieved by centrifugation (10 min, 2000 rpm) as opposed to filtration to save time and resources. Lipid weight was gravimetrically determined by placing triplicate aliquots of the lower chloroform phase into pre-weighed aluminium pans, allowing the samples to evaporate under a nitrogen atmosphere to prevent lipid oxidation at room temperature, then recording the weight on an analytical balance to the nearest 0.1 mg. The lower chloroform phase was removed using a glass Pasteur pipette, applying gentle positive-pressure while inserting through the upper phase.

Lipid data are expressed here as percent lipids ($P_L$):

$$P_L = \left(\frac{TL}{MD}\right) \times 100 \tag{4}$$

where $T_L$ = total lipids (g) and $M_D$ = total dried mass (g).

A subsample of 10 adults were analysed in triplicate to quantify within-individual variability. Values were later averaged. Owing to the amount of homogenate required, triplicates came from individuals >70 mm in total length (34 of the 68 adults). The selection of individuals for triplication was otherwise random. Standard deviations of triplicates were all <10% of the mean. Finally, a blank (made up of the aforementioned water, chloroform and methanol solution) was analysed alongside each sample to determine the presence of contaminants (i.e., lipid). None were found. A summary of all pertinent variables is provided (Table 1). Data are also available (Table S1).

**Table 1.** Descriptive statistics of the variables used in our analyses. S.D. = standard deviation; Fulton's *K* = Fulton's condition factor; *Ks* = Fulton's condition factor excluding stomach content and gonadal mass; *Kn* = Le Cren's relative condition factor. *n* = 68 and 25 for lethally sampled adult and juvenile *Cottus cognatus*, respectively. *n* = 937 and 509 for non-lethally sampled adult and juvenile *Cottus cognatus*, respectively.

| Data Type | Variable | Sex/Size | Mean ± S.D. | Range |
|---|---|---|---|---|
| Lethal | Total wet mass (g) | Adults | 4.0 ± 2.4 | 1.2–9.5 |
| | | Juveniles | 0.5 ± 0.2 | 0.2–1.0 |
| | Total Length (mm) | Adults | 71.0 ± 13.6 | 51–102 |
| | | Juveniles | 37.6 ± 4.2 | 29–45 |
| | Fulton's *K* | Adults | 1.0 ± 0.1 | 0.8–1.2 |
| | | Juveniles | 1.0 ± 0.1 | 0.8–1.2 |
| | *Ks* | Adults | 0.9 ± 0.1 | 0.7–1.2 |
| | | Juveniles | 0.9 ± 0.1 | 0.7–1.1 |
| | *Kn* | Adults | 0.9 ± 0.1 | 0.7–1.2 |
| | | Juveniles | 0.9 ± 0.1 | 0.7–1.1 |
| | Percent Lipid | Adults | 5.7 ± 1.8 | 2.4–11.3 |
| | | Juveniles | 4.3 ± 1.4 | 2.0–6.8 |
| Non-Lethal | Total wet mass (g) | Adults | 3.6 ± 2.0 | 1.1–16.2 |
| | | Juveniles | 0.6 ± 0.2 | 0.2–1.3 |
| | Total Length (mm) | Adults | 67.7 ± 10.8 | 50–112 |
| | | Juveniles | 38.7 ± 4.8 | 26–49 |

## 2.2. Statistical Analyses

The relationship between length and mass is often expressed as:

$$M = aL^b \tag{5}$$

where $M$ = total wet mass (g), $L$ = total length (mm), $a$ = constant equal to the intercept, and $b$ = allometric coefficient which, when equal to three indicates isometric growth, and any value substantially different than three indicates allometric growth [36].

The above equation was transformed by taking the natural logarithms ($\log_e$) of both sides:

$$\log_e(M) = \log_e(a) + b \log_e(L) \tag{6}$$

This allows the exponent $b$ and its 95% confidence interval to be estimated by simple linear regression. We did this for both adult and juvenile *C. cognatus* using non-lethal sample data collected each October from the same aforementioned sampling locations between 2013–2015 (specifically, individuals <50 mm were presumed to be juveniles and those ≥50 mm adults, as per Brasfield et al. [6]). Given the <25% overlap of confidence intervals [37] associated with each estimate of $b$ (Table 2), juvenile and adult non-lethal sample data were not pooled. Consequently, we produced two length-mass relationships to calculate $Kn$.

**Table 2.** Mass-length relationships for lethal and non-lethal samples, and their respective combinations. Coefficients of determination ($R^2$), root mean square error of the regression ($RMSE$), and the 95% confidence interval (C.I.) for each allometric coefficient ($b$) estimate are provided.

| Lethal/Non-Lethal | Maturity Status/Sex | Mass-Length Relationship | $n$ | C.I. for $b$ | $R^2$ | $RMSE$ |
|---|---|---|---|---|---|---|
| Non-lethal | Adult | $\log_e(M) = -11.97 + 3.12 \times \log_e(L)$ | 937 | 3.08–3.16 | 0.96 | 0.10 |
| | Juvenile | $\log_e(M) = -11.05 + 2.89 \times \log_e L$ | 509 | 2.81–2.96 | 0.92 | 0.11 |
| | Combined | $\log_e(M) = -11.51 + 3.01 \times \log_e(L)$ | 1446 | 2.99–3.03 | 0.99 | 0.10 |
| Lethal | Adult Females | $\log_e(M) = -11.66 + 3.03 \times \log_e(L)$ | 36 | 2.85–3.22 | 0.97 | 0.10 |
| | Adult Males | $\log_e(M) = -12.03 + 3.12 \times \log_e(L)$ | 28 | 2.85–3.39 | 0.95 | 0.11 |
| | Combined | $\log_e(M) = -11.83 + 3.07 \times \log_e(L)$ | 68 [1] | 2.94–3.21 | 0.97 | 0.10 |

[1] Includes four individuals that were not sexed.

The sex of non-lethally collected *C. cognatus* was not determined. However, there appears to be little difference in the allometric coefficients of adult male and female *C. cognatus* (Table 2). The sex of lethally sampled juveniles was also undetermined owing to a lack of a microscope in the field and an omission to inspect them on our return. We do not consider this a limitation given that differences in allometric coefficients are more likely to be apparent between adult female and male *C. cognatus*, for which we found no evidence. Consequently, data were only pooled by presumed maturity status. With regard to calculating $Kn$, we also explored pooling by river regulation type (i.e., regulated or unregulated), however, we found that this did not improve our ability to predict percent lipid [38].

General linear models were used to determine which indirect condition index best approximated percent lipid. Coefficients of determination ($R^2$) were calculated to report the proportion of variation explained. Also reported were 95% confidence intervals denoting the range of values estimated to encompass the 'true' linear regression line with a 95% probability [39]. Residual plots were examined to evaluate select assumptions of linear regression and whether any observations had undue influence on the analysis. Prediction precision was assessed by a combination of: (i) root mean square error ($RMSE$) (simply the average distance, in the units of the dependent variable, data points lie from the fitted regression line); (ii) 95% prediction intervals (estimated range of values within which future observations are likely to fall with a 95% probability given what has already been observed); and (iii) 'leave-one-out' cross validation (LOOCV). Cross-validation is where a data set is split into a training set and a testing set. The former is then used to predict the values of the latter. In LOOCV this is repeated as many times as there are data points, so that each data point is omitted and subsequently predicted using the remaining data. $R^2$ and $RMSE$ values are calculated for each iteration. These are then averaged to provide a single value (of which we refer to as LOOCV $R^2$ and LOOCV $RMSE$, respectively). The values that are predicted can also be regressed against observed values to visually evaluate bias and prediction error.

Akaike's information criterion, corrected for small sample size ($AIC_c$) [40], was used to compare five candidate models, specifically one for each condition index using percent lipid as the response, and an intercept only model (the purpose of the latter being to allow us to ask the question: do our condition index models better fit our data than a model with no independent variable?) The "best" model(s) for each maturity status were selected on the basis of their $\Delta AIC_c$, Akaike weight ($w_i$), and evidence ratio (ER), where: i) $\Delta AIC_c$ is the difference in $AIC_c$ scores between a model and the top model; ii) Akaike weight is analogous to the probability that out of the candidate models considered, a given model was the best approximating model; and iii) ER indicates how many times more likely the top model is at being the best approximating model than another [40]. As a general guide: models with $\Delta AIC_c$ values of < 2 are considered as good as the top model; models with $\Delta AIC_c$ values between 2 and 6 have moderate support and should therefore not be discounted; and for values > 6, model rejection can be considered (especially for values are >10) [40,41].

All statistical analyses were performed using R statistical software, version 3.4.3. Cross validation was conducted using the R package "caret". Assumptions of linear regression [39] were checked using the R package "gvlma". Significance of the associated tests was assessed at an α of <0.05.

## 3. Results

Mean percentage lipid values were 5.7 ± 1.8% (SD) for adults and 4.3 ± 1.4% for juveniles. The values of all individuals ranged from 2.0% to 11.3% (Table 2). Percent lipid was positively correlated with all condition indices in both adult and juvenile *C. cognatus*. Neither displayed perfect isometric growth (Table 2). Instead, adults tended toward positive allometric growth whereas juveniles displayed negative allometric growth. These trends were apparent in both lethal and non-lethal sample data, though only the confidence intervals of non-lethally sampled adults and juveniles did not include 3.0 (Table 2). Oddly, those of lethally sampled *C. cognatus* did, perhaps because of greater variation brought on by a smaller sample size. If we favour the non-lethal data then, and use 95% confidence intervals as our primary means of determining whether growth is isometric or allometric, we conclude there was a slight but significant proclivity for adult *C. cognatus* to become plumper as they grow in length, and for juveniles to elongate in form with increasing length.

### 3.1. Adults

Percent lipid was positively correlated with all three indices (Figure 1). Fulton's *K*, *Ks* and *Kn* performed equally well, accounting for 41%, 44%, and 42% of the variation in percent lipid, respectively (Table 3). Each model was further assessed by LOOCV. The resultant LOOCV $R^2$ values for the morphometric indices, while diminished, were not markedly so and remained somewhat large (0.37–0.39; Table 3) suggesting that predicted and observed observations did not substantially deviate from one another. The associated LOOCV *RMSE* values remained high (specifically, 1.37–1.39; Table 3).

High percent lipid values (i.e., ≥~8%) were consistently underestimated by the morphometric models (Figure 2a–c). Additionally, low percent lipid values (i.e., ≤~4%) seemed to tend toward being overestimated. In both instances, this was likely because the majority of individuals that contributed to each regression had mid-range values, driving predictions toward this range. The majority of high values were found to have relatively high leverage, thereby potentially distorting the outcome of the regressions. That said, the Cook's distance (which measures the influence of an observation when removed from the sample) of these observations were all <1 (data not shown) (as a general rule, values greater than 1 indicate highly influential observations [42,43]).

According to the aforementioned $\Delta AIC_c$ guidelines, the two models that included *Ks* and *Kn* received the most support (Table 3). Given that their $\Delta AIC_c$ values are both <2, they should be considered interchangeable. The Fulton's *K* model received moderate support (with values of <3).

RMSE values (Table 3) were all ~1.4% which, given the range of data analysed, indicates that the majority of observed values fell a fair distance from the regression line. Indeed, prediction precision seems poor across condition indices as reflected by the distance between prediction limits (Figure 1).

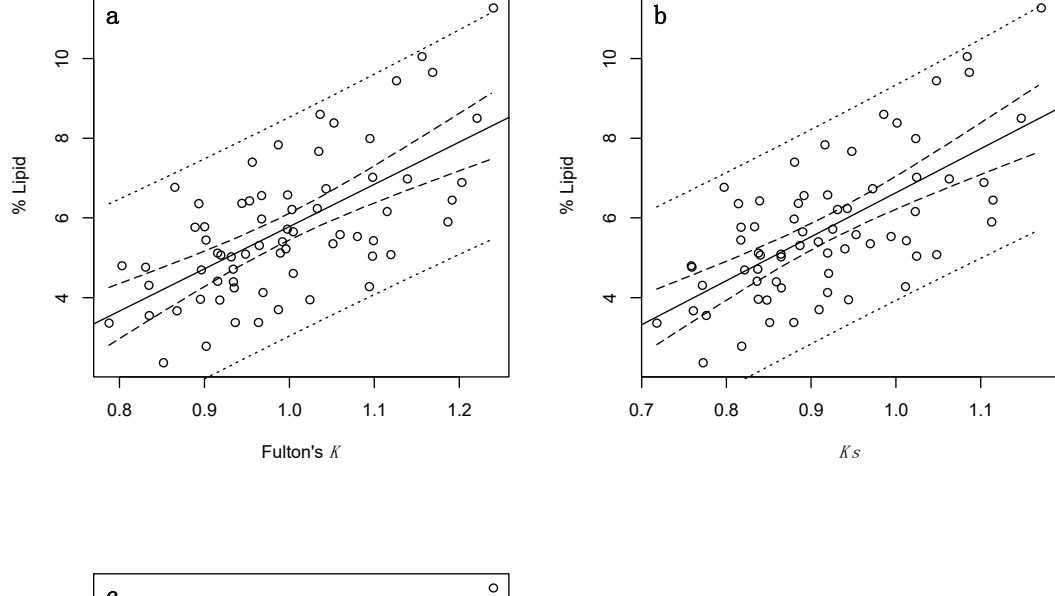

**Figure 1.** regressions of percent lipid and selected indirect condition indices for adult *Cottus cognatus* (*n* = 68), specifically: (**a**) Fulton's condition factor (Fulton's *K*); (**b**) Fulton's condition factor excluding stomach content and gonadal mass (*Ks*); and (**c**) Le Cren's relative condition factor (*Kn*). Solid lines represent linear regression lines; dotted lines denote 95% prediction intervals; and dashed lines denote 95% confidence intervals.

**Table 3.** linear models ranked to estimate which indirect condition index best approximates percent lipid in adult *Cottus cognatus* (*n* = 68). Fulton's *K* = Fulton's condition factor; *Ks* = Fulton's condition factor excluding stomach content and gonadal mass; *Kn* = Le Cren's relative condition factor; $R^2$ = coefficient of determination; *RMSE* = root mean square error; $AIC_c$ = bias-adjusted Akaike Information Criterion for small sample sizes; $\Delta AIC_c$ = $AIC_c$ of alternative model minus that of the top model; $w_i$ = Akaike weight; ER = evidence ratio); LOOCV $R^2$ = coefficient of determination from cross-validation; LOOCV *RMSE* = root mean square error from cross-validation. The coefficients of each predictor were significantly different from 0 ($p$ = <0.05).

| Candidate Model | $R^2$ | *RMSE* | $AIC_c$ | $\Delta AIC_c$ | $w_i$ | ER | LOOCV $R^2$ | LOOCV *RMSE* |
|---|---|---|---|---|---|---|---|---|
| *Ks* | 0.44 | 1.34 | 236.84 | 0.00 | 0.59 | - | 0.39 | 1.37 |
| *Kn* | 0.42 | 1.35 | 238.43 | 1.59 | 0.27 | 2.22 | 0.38 | 1.38 |
| Fulton's *K* | 0.41 | 1.37 | 239.65 | 2.81 | 0.14 | 4.07 | 0.37 | 1.39 |
| Intercept-only | - | 5.99 | 438.49 | 201.65 | 0.00 | $6.12^{43}$ | - | - |

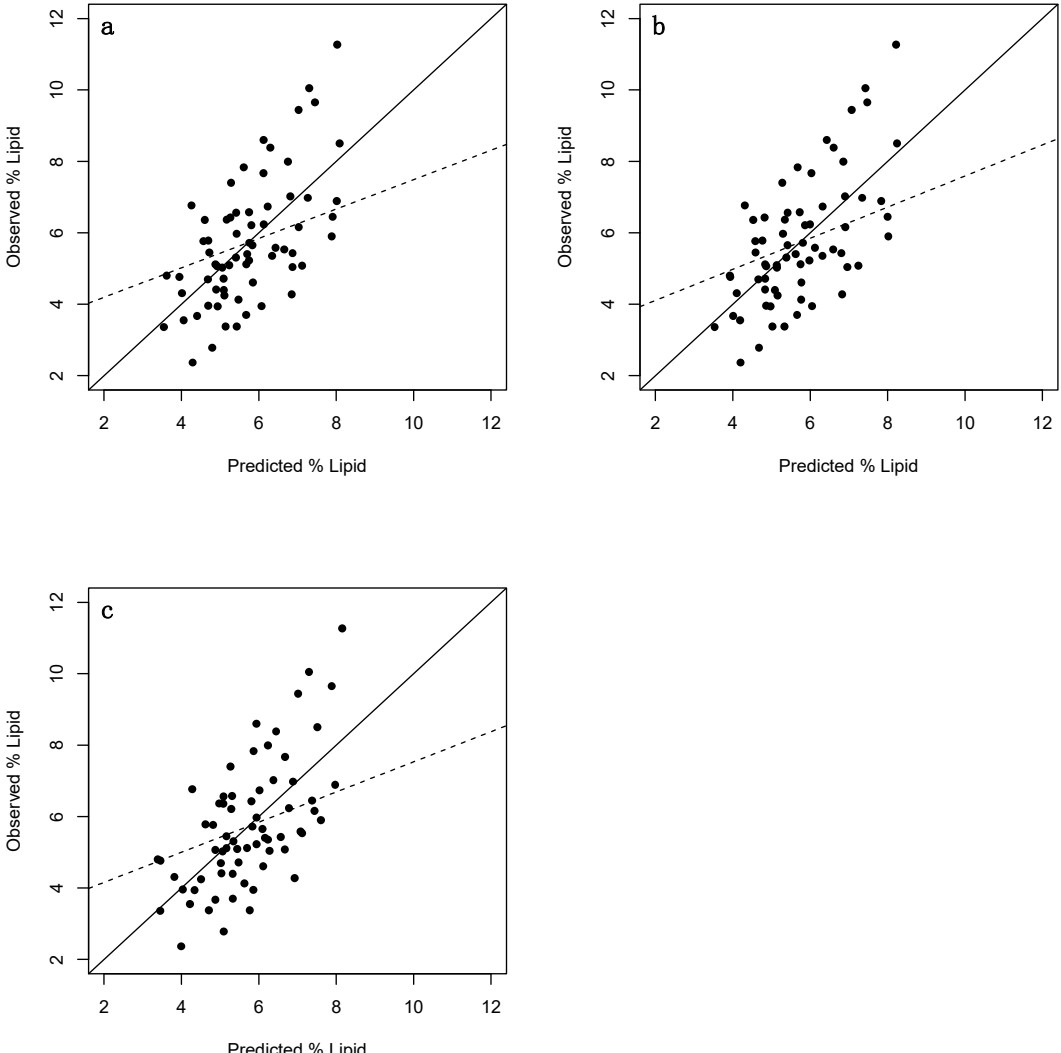

**Figure 2.** plots of observed vs. predicted percent lipid values of adult *Cottus cognatus* (*n* = 68). (**a**) = Fulton's condition factor; (**b**) = Fulton's condition factor excluding stomach content and gonadal mass; (**c**) = Le Cren's relative condition factor. Solid lines have a slope of 1 and an intercept of 0 thereby representing the ideal scenario where predicted = observed. Dashed lines denote represent linear regression lines.

*3.2. Juveniles*

Juvenile percent lipid values were also positively correlated with all three indices (Figure 3). Interestingly, each index accounted for slightly more variation than their adult counterparts (specifically, 48, 51, and 52% for Fulton's *K*, *Ks* and *Kn*, respectively (Table 4)). The *RMSE* values associated with the juvenile data were also lower than those of the adults (at ~ 1%), and the prediction intervals slightly narrower, though not sufficiently so to alleviate concerns regarding prediction precision. LOOCV $R^2$ and LOOCV *RMSE* values were largely similar, ranging from 0.41–0.45 and 1.03–1.06%, respectively. Furthermore, a moderate proportion of observed values were in agreement with those predicted by LOOCV (Figure 4a–c), and no unusual residual observations were identified, though it should still be noted that a relatively small number of juveniles (*n* = 25) were analysed.

While the *Kn* model received the most support, there is near equal support for the *Ks* model as reflected in the $\Delta$AIC$_c$ and $w_i$ values, of which were 0.00 and 0.11, and 0.42 and 0.40, respectively (Table 4). Fulton's *K* also received considerable support with $\Delta$AIC$_c$ and $w_i$ values of 1.75 and 0.18,

respectively, and should therefore not be discounted. Indeed, the *Kn* model was only 2.27 times more likely to be the best approximating model than the Fulton's *K* model.

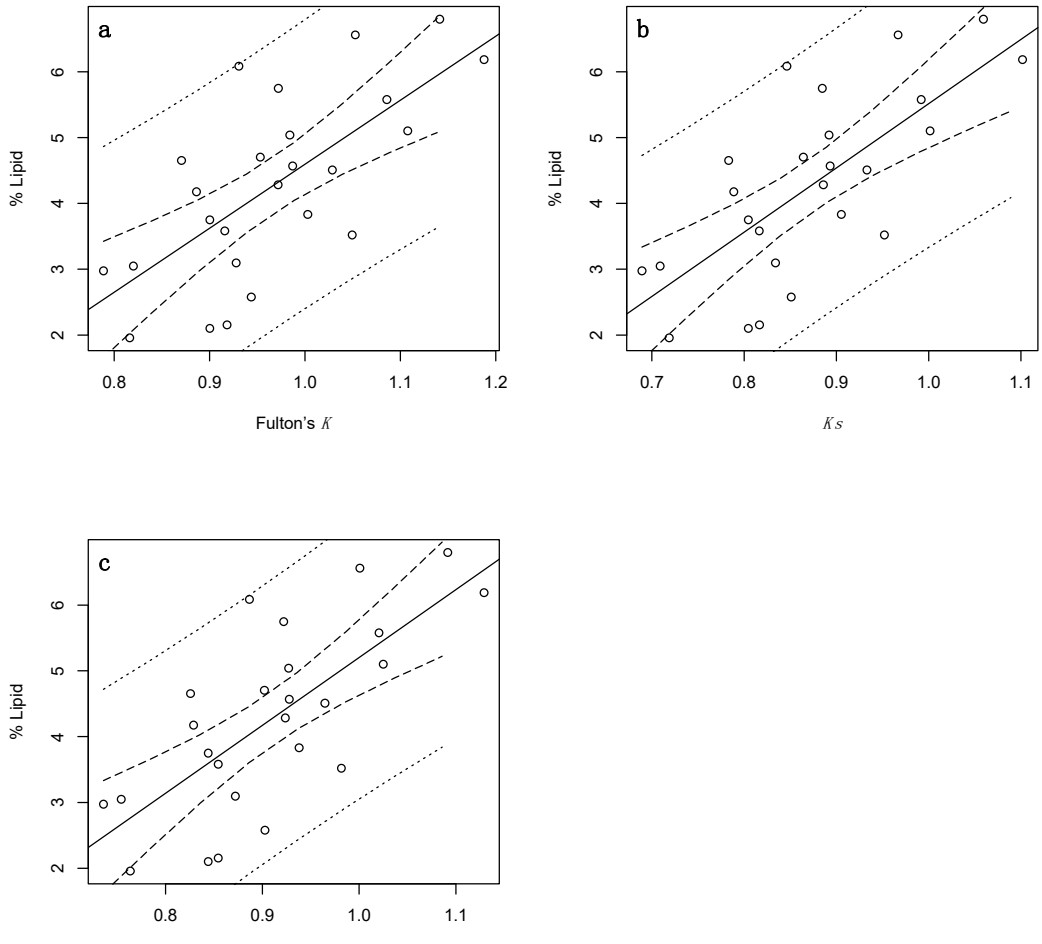

**Figure 3.** regressions of percent lipid and selected indirect condition indices for juvenile *Cottus cognatus* (*n* = 25), specifically: (**a**) = Fulton's condition factor (Fulton's *K*); (**b**) = Fulton's condition factor excluding stomach content and gonadal mass (*Ks*); and (**c**) = Le Cren's relative condition factor (*Kn*). Solid lines represent linear regression lines; dotted lines denote 95% prediction intervals; and dashed lines denote 95% confidence intervals.

**Table 4.** models ranked to estimate which indirect condition index best approximates percent lipid in juvenile *Cottus cognatus* (*n* = 25). *PL* = percent lipid; Fulton's *K* = Fulton's condition factor; *Ks* = Fulton's condition factor excluding stomach content and gonadal mass; *Kn* = Le Cren's relative condition factor; $R^2$ = coefficient of determination; *RMSE* = root mean square error of the regression; $AIC_c$ = bias-adjusted Akaike Information Criterion for small sample sizes; $\Delta AIC_c$ = $AIC_c$ of alternative model minus that of the top model; $w_i$ = Akaike weight; ER = evidence ratio); LOOCV $R^2$ = coefficient of determination from cross-validation; LOOCV *RMSE* = root mean square error from cross-validation. The coefficients of each predictor were significantly different from 0 (*p* = <0.05).

| Candidate Model | $R^2$ | *RMSE* | $AIC_c$ | $\Delta AIC_c$ | $w_i$ | ER | LOOCV $R^2$ | LOOCV *RMSE* |
|---|---|---|---|---|---|---|---|---|
| *Kn* | 0.52 | 1.00 | 76.13 | 0.00 | 0.42 | - | 0.45 | 1.03 |
| *Ks* | 0.51 | 1.01 | 76.24 | 0.11 | 0.40 | 1.06 | 0.45 | 1.03 |
| Fulton's *K* | 0.48 | 1.04 | 77.88 | 1.75 | 0.18 | 2.27 | 0.41 | 1.06 |
| Intercept-only | - | 4.48 | 148.12 | 71.99 | 0.00 | $4.06^{15}$ | - | - |

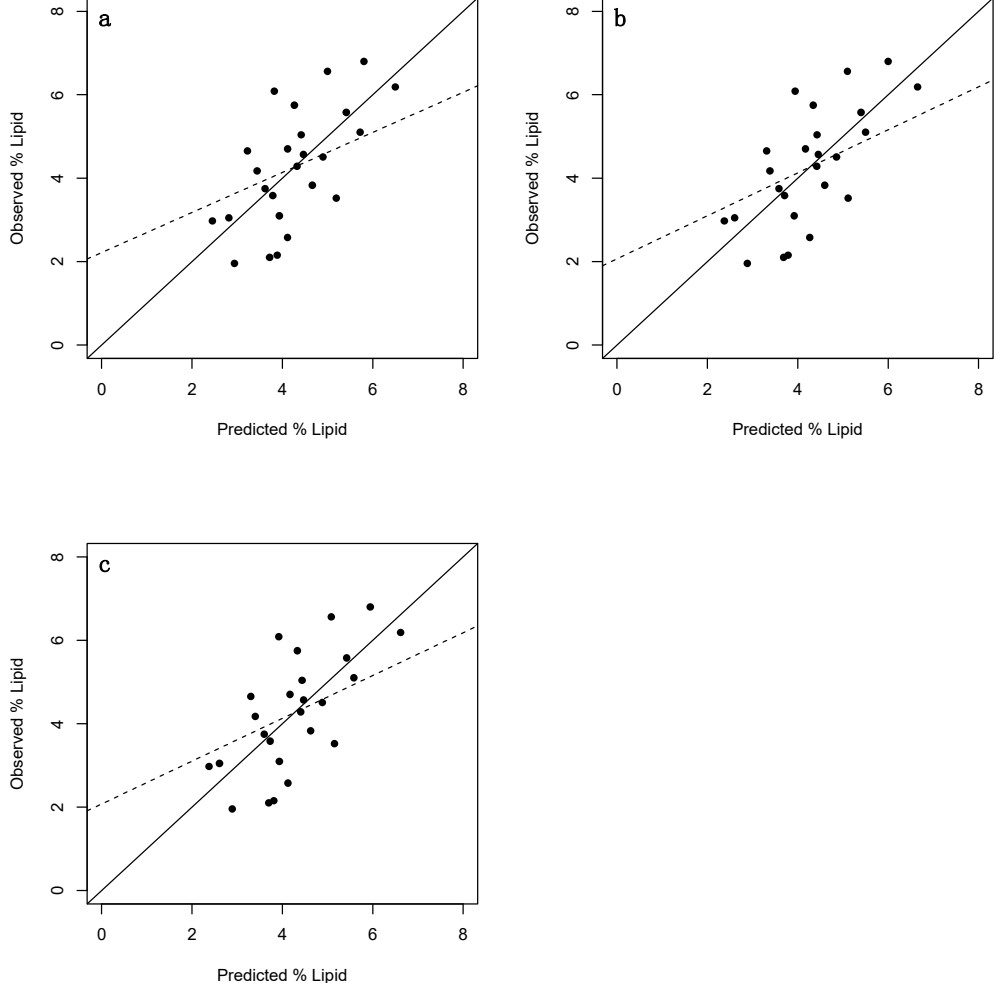

**Figure 4.** plots of observed vs. predicted percent lipid values of juvenile *Cottus cognatus* (*n* = 25). (**a**) = Fulton's condition factor; (**b**) = Fulton's condition factor excluding stomach content and gonadal mass; (**c**) = Le Cren's relative condition factor. Solid lines have a slope of 1 and an intercept of 0 thereby representing the ideal scenario where predicted = observed. Dashed lines denote represent linear regression lines.

## 4. Discussion

Condition indices derived from length-mass data provide a rapid and potentially non-lethal means of assessing the relative plumpness of fish. If the assumption that plumpness is strongly related to lipid stores (specifically triglycerides, a subgroup of lipids) is valid, they can additionally be used as surrogates of energy storage. While there is evidence to suggest this assumption may be relevant for some species, lipids have been found to be poorly correlated with condition factor in others [44]. Results from this study suggest there was a positive correlation between each of the selected condition indices and percent lipid. The degree to which they were related, however, varied between indices.

Of the three morphometric conditions tested, *Ks* and *Kn* performed the "best" and Fulton's *K* performed the "worst", though not to the degree we expected. Indeed, relative to *Kn*, Fulton's *K* performed well among both juvenile and adult populations despite the fact that they displayed negative and positive allometric growth respectively, thereby violating the assumption of isometric growth. Allometric coefficients derived in this study (Table 2) were, however, not unusual. Carlander [45], for example, found that for a variety of freshwater fish species, values typically fall between 2.5 and 3.5. And given that *C. cognatus* displayed only slight allometry, it is unsurprising that *Kn* performed

only marginally better. This suggests that in the absence of historical data, Fulton's *K* may be used as substitute for *Kn*.

Also surprising was how negligible the confounding effects of stomach contents and gonads were. Indeed, although *Ks* performed better than Fulton's *K*, the improvement was marginal and *Kn* performed just as well (if not better, at least with regard to juveniles). While this result may not be common [17,30], it is not unique. Le Cren [22], for example, concluded that the stomach contents of perch *Perca fluviatilis* (Linnaeus 1758) were "not important", contributing at most 2% of the body weight. With regard to gonadal development, male *C. cognatus* were likely undergoing recrudescence at the time of sampling [6]. One might expect this to reduce the predictive utility of Fulton's *K* and *Kn*, but male gonad investment is likely minimal relative to their body size [6]. Further, we did not see a substantial difference between male and female *C. cognatus* in this study. Overall, this suggests there is little benefit in using somatic mass over non-lethally obtained wet mass, at least for the time of year sampled.

Previous studies that investigated the use of condition indices as surrogates for lipid content in other fishes primarily use $R^2$ as the means by which to assess utility. And while no specific value is used as a benchmark, values > 0.4–0.5 result in the associated index being recommended (for example, see: Herbinger and Friars [23] for Atlantic salmon; Mozsár et al. [46] for Amur sleeper *Perccottus glenii* (Dybowski 1877) and common rudd *Scardinius erythrophthalmus* (Linnaeus 1758); Schloesser and Fabrizio [47] for summer flounder *Paralichthys dentatus* (Linnaeus 1766) and Atlantic croaker *Micropogonias undulatus* (Linnaeus 1766); and Brosset et al. [48] for European anchovy *Engraulis encrasicolus* (Linnaeus 1758), the European pilchard *Sardina pilchardus* (Walbaum 1792) and the European sprat *Sprattus sprattus* (Linnaeus 1758)). Conversely, studies with values of ≤ 0.25 do not seem to result in an index being recommended (for example, see: Davidson and Marshall [33] and McPherson et al. [49] for North Sea herring *Clupea harengus* (Linnaeus 1758); Jonas et al. [50] for muskellunge *Esox masquinongy* (Mitchill 1824); Schloesser and Fabrizio [47] for striped bass *Morone saxatilis* (Walbaum 1792); Mozsár et al. [49] for pumpkinseed *Lepomis gibbosus* (Linnaeus 1758); and Chellappa et al. [51] for male three-spined stickleback *Gasterosteus aculeatus* (Linnaeus 1758).

Therefore, while the $R^2$ values reported in this study are unremarkable, especially in isolation, they are at least equivalent to the $R^2$ values of studies that a number of researchers continue to cite to justify their use of morphometric indices.

An arguably more useful metric is *RMSE*, and here the values are somewhat disappointing. LOOCV *RMSE* values ranged from 1.37 to 1.39 for adults and 1.03 to 1.06 for juveniles. In the context of the values sampled this seems high (1.06 represents a fifth of the range of juvenile values, for example). The similarity in predictive utility among the indices suggests that the majority of variation left unexplained is associated with something that all morphometric indices share. The shape of fish can, for example, be associated with changes in constituents other than lipid, such as protein [52,53].

Total body lipid content (expressed as percent dry weight) of *C. cognatus* had a combined mean of 5.3 ± 1.8% (SD). This is comparable to *C. cognatus* collected from the Great Lakes, in which values were found to vary from 5.0% to 8.3% [54]. Conversely, our mean was noticeably higher than that of Bennett et al. [55] whose riverine *C. cognatus* from northern Saskatchewan had a mean of 2.2% (though it should be noted some of the individuals that contributed to this value were collected in May).

## 5. Conclusions

Condition indices are commonly used but rarely defined or validated. In their review, Hayes and Shonkwiler [16] asserted that " ... often the intended biological meaning of a condition index is nebulous ... and their validity is not verified". For *C. cognatus* at least, there was a moderate degree of evidence to suggest that morphometric-based condition indices are reliable predictors of percent lipid. The range of percent lipid values measured in this study was also rather high, thereby increasing the likelihood that environmental stressors will result in noticeable changes in percent lipid [46].

Given the proliferation of *C. cognatus* as an environmental indicator species and the logistical difficulties associated with routinely measuring fat content directly, this is welcome news.

Although *Kn* and *Ks* indices performed consistently well with both juvenile and adult data, we recommend the use of *Kn* because it does not require that fish be killed. If the data necessary to calculate *Kn* have not already been collected then Fulton's *K* is a suitable substitute. That said, our recommendations come with strong caveats: owing to the need to divide individuals by maturity status, our sample sizes, for juveniles in particular, were somewhat small. Small sample sizes decrease the likelihood of detecting aberrant measurements [30]. Furthermore, all individuals came from one relatively small sampling area and from one time of year, thereby spatially and temporally limiting the applicability of our findings. We would not, for example, recommend the use of such indices immediately prior to spawning (which, in New Brunswick, occurs in mid to late May [7]). We also acknowledge that we did not measure triglycerides exclusively, but instead used the whole individual and thereby measured both storage and structural lipids. While this is a limitation to our work, Bennett et al. [55] reported a high degree of correlation between triglycerides and total body lipids in *C. cognatus*. Moreover, the Bligh and Dyer [19] method is still widely used [56].

Nevertheless, we believe such morphometric indices are sufficient for predicting fat content in *C. cognatus* to a degree that likely suits the needs of most researchers. Although newer, sometimes promising, non-lethal methods for assessing fat content in fish have emerged [48,57,58], their apparent lack of adoption outside of studies examining their efficacy suggests that traditional methods are not going away anytime soon.

**Supplementary Materials:** The following are available online at http://www.mdpi.com/1424-2818/11/5/71/s1, Table S1: Adult and juvenile: percent lipid; Fulton's K; Kn; and Ks data used in our analyses.

**Author Contributions:** Conceptualization, A.R.H.; Data curation, A.R.H.; Formal analysis, A.R.H.; Funding acquisition, M.A.G. and R.A.C.; Investigation, A.R.H. and S.C.N.; Methodology, A.R.H. and S.C.N.; Project administration, M.A.G. and R.A.C.; Supervision, A.R.H. and R.A.C.; Writing—original draft, A.R.H.; Writing—review & editing, M.A.G. and R.A.C.

**Funding:** This work was supported by the Canada Research Chairs Program (#CRC 950-200379), a Natural Sciences and Engineering Research Council (NSERC) Discovery Grant (#RGPIN 203894-2005), an NSERC HydroNet Strategic Network Grant (#Montreal/NETGP 370899/HydroNet & #Montreal/NETGP 370899/Egg survival), the New Brunswick Innovation Foundation (NBIF) (#RAI 2015-054), and the UNB Work-Study Program (#NA).

**Acknowledgments:** We thank H. A. Burke, S. A. McGeachy and M. Savoie from UNB for their assistance with lipid extraction; A. D. Annis and R. D. Voisine for assisting with sample collection; and the three anonymous reviewers whose comments greatly improved this article.

**Conflicts of Interest:** The authors declare no conflict of interest. The funders had no role in the design of the study; in the collection, analyses, or interpretation of data; in the writing of the manuscript, or in the decision to publish the results.

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
