# Peer review of "Utility of Condition Indices as Predictors of Lipid Content in Slimy Sculpin (Cottus cognatus)"

_diversity, doi:10.3390/d11050071_

Round 1
Reviewer 1 Report
Review of Diversity 479221
This paper examines whether lipid content of slimy sculpin can be predicted with 4 different morphometric indices, some of which do not require lethal sampling. Assuming the results can be generalized to slimy sculpin anywhere, the information will be useful to anyone hoping to use sculpin as an indicator species, or to those interested in how morphometric indices compare to other physiological indices, such as lipid content.
Overall, the paper is well done. There are some writing aspects that could be redone more clearly. The overall conclusion that the morphometric indices are a reliable predictor of lipids is probably overstated given the high variability around each regression line.
Some specific comments are below.
Line 97 It is a bit confusing here to mention lethal and non-lethal sampling without any prior context.
Line 142 were you able to get a fine enough homogenate with a mortar and pestle to do the analysis for consistent replicates?
Line 147 what is the reasoning behind non replicated vs replicated samples in each run?
Table 2 I would add b to the column lable CI so we know the CI is for the slope
Line 255 It is confusing to discuss the results for H here right after discussing the non-transformed results for K, Ks, and Kn, because the R2 values given in line 254 aren’t the ones that are comparable to the R2 for H, which is not part of Table 3.
Table 3 legend—delete information about H, it is not in the table. Line 267 description of R2 and RMSE should come earlier in the legend, right after Kn so it goes in order left to right across the column labels (same for Table 4, and Table 5 legent)
Line 277 Again, H is not part of the results provided in Table 3, so this sentence seems out of place when it is provided along with the non-transformed analysis.
Figure 2. It would be useful to see the regression for the predicted vs observed points, and provided a slope, to see how the line deviates from the 1:1 line, and how the slope deviates from 1. Same comment for Figure 3.
Table 5 delete Hi from legend, it is not in the table
Discussion, although separating ideas with a semicolon is probably not technically wrong, to me at least, I found the frequent use of semi colons in the paper somewhat distracting, and wonder if using periods and new sentences might make things clearer.
Discussion Many of the paragraphs lack a clear topic or argument.
Line 368 Sometimes you have references to something, “these, that” and it is not entirely clear what you are referring to, so make sure it is clear what you are referring to in each sentence.
Line 393 I’m not sure this would indicate differing lipid requirements as much as differing “health” among these different populations, which is the premise for the need of this study.
Line 404 although R2 of 0.41-0.44 is probably high for this type of analysis, it is not really that high and indicates a lot of variation around the predicted values.
Line 410 Not clear what “These” refers to
Line 416-417 What did you expect given the range of lipid values that were sampled and why?
Line 422 not sure what “their” refers to
Line 424-440 Not sure what the idea of this paragraph is, there are a lot of ideas in here. It is not clear why the variation should be attributed to sampling. There are many factors that affect weight of a fish other than lipids, so it should not be all that surprising that the relationships between condition factors and lipids were not stronger. Considering all the potential factors affecting a fish wet weight other than lipids, don’t see why using lipids vs triglycerides would necessarily be an issue. Is there any evidence that it would?
Line 441 But above you noted a study that showed higher lipids in fall. Further, if the point is to see if there is a relationship between condition indices and lipids, then when you sample should not be an issue, unless the relationship changes seasonally, which would decrease the utility of your study.
Line 462-464 I’m not sure I follow the logic of this sentence, why would you expect this?
Line 469-470 This sentence is hard to follow, it should be rewritten
Line 471 I’m not sure “dependable” is the right word. Although it is clear that as condition index values increase, lipids increase, but there is still a lot of variation in these relationships that one might not want to use them as predictive, depending on their study requirements.
Line 480 what do you mean by “fat”, triglycerides?
Author Response
Response to Reviewer 1 Comments
Point 1: Overall, the paper is well done. There are some writing aspects that could be redone more clearly. The overall conclusion that the morphometric indices are a reliable predictor of lipids is probably overstated given the high variability around each regression line.
Response 1: Please see ‘response 14’
Point 2: Line 97 It is a bit confusing here to mention lethal and non-lethal sampling without any prior context.
Response 2: Lines 102-107 modified to provide additional context
Point 3: Line 142 were you able to get a fine enough homogenate with a mortar and pestle to do the analysis for consistent replicates?
Response 3: Yes. The use of a mortar and pestle for such purposes is still quite common. Moreover, samples that we ran in triplicate did not deviate substantially from one another (please see lines 148-152).
Point 4: Table 2 I would add b to the column label CI so we know the CI is for the slope
Response 4: Added “for b” to the column referred to in Table 2
Point 5: Line 255 It is confusing to discuss the results for H here right after discussing the non-transformed results for K, Ks, and Kn, because the R2 values given in line 254 aren’t the ones that are comparable to the R2 for H, which is not part of Table 3.
Response 5: Section “3.1 Adults” rearranged and reworded. Please note, data for Hi now excluded owing to the comments of another reviewer (whom rightly pointed out that the readability of the balance used was not sufficient).
Point 6: Table 3 legend—delete information about H, it is not in the table. Line 267 description of R2 and RMSE should come earlier in the legend, right after Kn so it goes in order left to right across the column labels (same for Table 4, and Table 5 legend)
Response 6: Have adjusted order of information about R2 and RMSE in each Table description.
Point 7: Line 277 Again, H is not part of the results provided in Table 3, so this sentence seems out of place when it is provided along with the non-transformed analysis.
Response 7: Please see ‘response 6’
Point 8: Figure 2. It would be useful to see the regression for the predicted vs observed points, and provided a slope, to see how the line deviates from the 1:1 line, and how the slope deviates from 1. Same comment for Figure 3.
Response 8: Added linear regression lines to Figures 2 and 4. Associated R2 values can be found in Tables 2 and 3.
Point 9: Table 5 delete Hi from legend, it is not in the table
Response 9: Information about Hi removed
Point 10: Discussion, although separating ideas with a semicolon is probably not technically wrong, to me at least, I found the frequent use of semi colons in the paper somewhat distracting, and wonder if using periods and new sentences might make things clearer.
Response 10: The vast majority of semi-colons have been removed. Please let us know of any glaring examples that remain.
Point 11: Discussion. Many of the paragraphs lack a clear topic or argument.
Response 11: We have reworded and restructured the discussion. Paragraph 368-373 still seems somewhat out of place, but we feel it is important to compare our percent lipid values against those of other studies.
Point 12: Line 368 Sometimes you have references to something, “these, that” and it is not entirely clear what you are referring to, so make sure it is clear what you are referring to in each sentence.
Response 12: Edited discussion and conclusion to replace “these” and “that” with something more specific. Please let us know if any examples remain.
Point 13: Line 393 I’m not sure this would indicate differing lipid requirements as much as differing “health” among these different populations, which is the premise for the need of this study.
Response 13: The line referred to has been deleted
Point 14: Line 404 although R2 of 0.41-0.44 is probably high for this type of analysis, it is not really that high and indicates a lot of variation around the predicted values.
Response 14: Lines 345-361. While we agree our results are, in isolation, unremarkable, compared to what researchers who use condition indices seem to think is acceptable (based on citations), we believe they are adequate. That said, we have reworded our discussion to temper our recommendations somewhat and better highlight our study’s limitations.
Point 15: Line 410 Not clear what “These” refers to
Response 15: see ‘response 12’
Point 16: Line 416-417 What did you expect given the range of lipid values that were sampled and why?
Response 16: Reworded lines 362-367 to better indicate why these values matter relative to the range of percent lipid values recorded. As to what we expected (or would have liked to see), that’s a difficult question to answer because, like R2, it’s subjective. And while there seems to be at least some consensus, albeit unacknowledged, regarding what is and isn’t an acceptable R2 value, RMSE is rarely reported or discussed.
Point 17: Line 422 not sure what “their” refers to
Response 17: Associated sentence has been reworded
Point 18: Line 424-440 Not sure what the idea of this paragraph is, there are a lot of ideas in here. It is not clear why the variation should be attributed to sampling. There are many factors that affect weight of a fish other than lipids, so it should not be all that surprising that the relationships between condition factors and lipids were not stronger. Considering all the potential factors affecting a fish wet weight other than lipids, don’t see why using lipids vs triglycerides would necessarily be an issue. Is there any evidence that it would?
Response 18: Not that we are aware of. We have, as a result, removed that paragraph from the manuscript.
Point 19: Line 441 But above you noted a study that showed higher lipids in fall. Further, if the point is to see if there is a relationship between condition indices and lipids, then when you sample should not be an issue, unless the relationship changes seasonally, which would decrease the utility of your study.
Response 19: True. We have removed that line from the manuscript. The relationship between condition and total lipids is likely to change (for females especially) prior to the spawning season (~ March through to May) but the majority of environmental assessment sampling occurs in the fall.
Point 20: Line 462-464 I’m not sure I follow the logic of this sentence, why would you expect this?
Response 20: A more likely explanation is that there is some, albeit extremely minor, gonadal development in adults as opposed to none at all in juveniles. On review, this sentence doesn’t add anything of value to the manuscript and so has been removed.
Point 21: Line 469-470 This sentence is hard to follow, it should be rewritten
Response 21: The line in question has been shortened and reworded.
Point 22: Line 471 I’m not sure “dependable” is the right word. Although it is clear that as condition index values increase, lipids increase, but there is still a lot of variation in these relationships that one might not want to use them as predictive, depending on their study requirements.
Response 22: dependable has been removed. We have also reworded the conclusion as discussed in response to previous points so as not to exaggerate the applicability of our results.
Point 23: Line 480 what do you mean by “fat”, triglycerides?
Response 23: We do. We see that this sentence is potentially confusing so have reworded the latter part of the conclusion (from line 392 on).
Thank you for your comments

Reviewer 2 Report
General comments:
The study by Hards et al., presents data on slimy sculpin body condition measured with different indices. They compared the results obtained with four different indirect body condition indices with total lipid content. This topic is interesting as studies validating body condition indices are scarce while these indices are frequently used. Data are new, statistical demarche is clear but I have some concerns that need to be considered before I can declare the manuscript suitable for publication.
My main concern is the lack of discussion about some aspects essential of this work. After the line 375, you should develop on your idea about the slight differences between K, Kn and Ks. The slight difference of Kn in comparison to K and Ks is normal when you look at the a and b coefficient of the non linear relationship between length and weight. As b is not far from 3 which is used in K and Ks computation, this is normal that Kn and K was close. This suggest that for species with slight allometric growth, Kn only slightly improve your description of fish body condition.
Similarly, please stress on the fact that do not considering somatic weight (Ks) do not improve very much the relationship with total lipid content. This is not true for all species and is really relevant in your case because it argues that conduct non-lethal sampling (K or Kn with total weight) will allow you to have a reliable index of condition. This is really interesting as in general somatic weight or gutted weight give significantly better results.
Another major comment is about your total lipid content. Why did you used the whole individual and not only the fillet (i.e. muscle). In general, triglycerides are mainly either in muscle or in liver. So by taking the whole individual, you are measuring a lot of sterols, which are structure lipids. I do not request to redo analyses, but as it is a limitation, you should discuss this point in your manuscript (and maybe try to measure next time TAG and sterols separately).
In the introduction, you should add after the line 39 some example about the use of fish body condition. In the current state of the introduction, I understand what is body condition and the limitations, but I do not see any potential about these measurements. I know for example that fish body condition helped to understand predation pressure on larvae (Hoey and McCormick, 2004) , small pelagic fish dynamics (Brosset et al., 2017), fish fecundity (Brosset et al., 2016, Mion et al., 2018). You should add these exemple as you will be able to show what are the possibilities of research once you will have defined the reliability of your body condition indices.
My last major concern is about the lack of some essential references about fish body condition in the manuscript. Some studies should be used to enhance this study as similar conclusions can be made. This is important as this type of study is really important to know what you measure with your indirect index of condition (e.g. Fulton …). Please consider and add the references cited above and below in the minor comments (the references are listed at the end of the review) to strengthened the introduction and the discussion by clearly identifying possibilities offered with reliable non-lethal condition indices.
Minor points:
In the lines 33 and 39, you may add the essential book on fish condition (Lloret et al., 2013) in addition to your references 13 and 17.
The citation 20 in the line 48 is not adapted here. The definition of Fulton index is in Fulton, 1904.
In the line 55, in addition to your reference 22, you should add a very similar study to yours (Brosset et al., 2015). They follow the same framework that you used and may also serve to enhance your results with the similarities in the results in the discussion (add it also in the line 404 to 410 as it is a good example of what you show with your study).
References used in the review:
Brosset, P., Fromentin, J. M., Ménard, F., Pernet, F., Bourdeix, J. H., Bigot, J. L., ... & Saraux, C. (2015). Measurement and analysis of small pelagic fish condition: a suitable method for rapid evaluation in the field. Journal of experimental marine biology and ecology, 462, 90-97.
Brosset, P., Lloret, J., Muñoz, M., Fauvel, C., Van Beveren, E., Marques, V., ... & Saraux, C. (2016). Body reserves mediate trade-offs between life-history traits: new insights from small pelagic fish reproduction. Royal Society open science, 3(10), 160202.
Brosset, P., Fromentin, J. M., Van Beveren, E., Lloret, J., Marques, V., Basilone, G., ... & De Felice, A. (2017). Spatio-temporal patterns and environmental controls of small pelagic fish body condition from contrasted Mediterranean areas. Progress in oceanography, 151, 149-162.
Fulton. 1904. The rate of growth of fishes. 22nd Annual Report of the Fishery Board of Scotland 1904 (3):141-241.
Hoey, A. S., & McCormick, M. I. (2004). Selective predation for low body condition at the larval-juvenile transition of a coral reef fish. Oecologia, 139(1), 23-29.
Lloret, J., Shulman, G., & Love, R. M. (2013). Condition and health indicators of exploited marine fishes. John Wiley & Sons.
Mion, M., Thorsen, A., Vitale, F., Dierking, J., Herrmann, J. P., Huwer, B., ... & Casini, M. (2018). Effect of fish length and nutritional condition on the fecundity of distressed Atlantic cod Gadus morhua from the Baltic Sea. Journal of fish biology, 92(4), 1016-1034.
Author Response
Response to Reviewer 2 Comments
Point 1: My main concern is the lack of discussion about some aspects essential of this work. After the line 375, you should develop on your idea about the slight differences between K, Kn and Ks. The slight difference of Kn in comparison to K and Ks is normal when you look at the a and b coefficient of the non linear relationship between length and weight. As b is not far from 3 which is used in K and Ks computation, this is normal that Kn and K was close. This suggest that for species with slight allometric growth, Kn only slightly improve your description of fish body condition.
Response 1: Lines 325-333. We have expanded the comparison between Fulton’s K and Kn.
Point 2: Similarly, please stress on the fact that do not considering somatic weight (Ks) do not improve very much the relationship with total lipid content. This is not true for all species and is really relevant in your case because it argues that conduct non-lethal sampling (K or Kn with total weight) will allow you to have a reliable index of condition. This is really interesting as in general somatic weight or gutted weight give significantly better results.
Response 2: Lines 334-344. We also have expanded on the comparison between Ks and Fulton’s K and Kn.
Point 3: Another major comment is about your total lipid content. Why did you used the whole individual and not only the fillet (i.e. muscle). In general, triglycerides are mainly either in muscle or in liver. So by taking the whole individual, you are measuring a lot of sterols, which are structure lipids. I do not request to redo analyses, but as it is a limitation, you should discuss this point in your manuscript (and maybe try to measure next time TAG and sterols separately).
Response 3: We now discuss triglycerides in both the introduction (lines 34-35) and discussion (lines 292-396). Although we recognise that measuring triglycerides exclusively would have been preferred, the use of the Bligh and Dyer method is still widely used. Further, Bennett et al. [55] found that total lipid was highly correlated with triglycerides in C. cognatus.
Point 4: In the introduction, you should add after the line 39 some example about the use of fish body condition. In the current state of the introduction, I understand what is body condition and the limitations, but I do not see any potential about these measurements. I know for example that fish body condition helped to understand predation pressure on larvae (Hoey and McCormick, 2004) , small pelagic fish dynamics (Brosset et al., 2017), fish fecundity (Brosset et al., 2016, Mion et al., 2018). You should add these example as you will be able to show what are the possibilities of research once you will have defined the reliability of your body condition indices.
Response 4: Agreed, and thank you for the suggestions. Please see lines 59-62.
Point 5: My last major concern is about the lack of some essential references about fish body condition in the manuscript. Some studies should be used to enhance this study as similar conclusions can be made. This is important as this type of study is really important to know what you measure with your indirect index of condition (e.g. Fulton …). Please consider and add the references cited above and below in the minor comments (the references are listed at the end of the review) to strengthen the introduction and the discussion by clearly identifying possibilities offered with reliable non-lethal condition indices.
Response 5: We have added the references suggested (please see lines 345-351).
Point 6: In the lines 33 and 39, you may add the essential book on fish condition (Lloret et al., 2013) in addition to your references 13 and 17.
Response 6: We have added the suggested reference. Thank you for the recommendation.
Point 7: The citation 20 in the line 48 is not adapted here. The definition of Fulton index is in Fulton, 1904.
Response 7: The history behind this one is rather interesting: The origin of the name “Fulton’s condition factor” and the first attribution of the equation to Fulton was actually by Ricker (as outlined here: https://pdfs.semanticscholar.org/31b0/e2b909bf7a0d51f0eea815572b352400b376.pdf). The book by Lloret et al. follows this suggestion. Regardless, we have also included Fulton’s original paper (please see reference [31]).
Point 8: In the line 55, in addition to your reference 22, you should add a very similar study to yours (Brosset et al., 2015). They follow the same framework that you used and may also serve to enhance your results with the similarities in the results in the discussion (add it also in the line 404 to 410 as it is a good example of what you show with your study).
Response 8: We have added the paper suggested to both the introduction and discussion.
Thank you for your comments.

Reviewer 3 Report
Overall comments
This paper compared the utility of four morphometric indices as predictors of lipid content in Slimy Sculpin. Overall I liked the paper and found it quite informative. However there are several serious issues that should be addressed before I would consider the paper publishable. First, I am concerned with the validity of the HSI analysis because the scale used in the field was so insensitive (+- 0.01 g). The authors report in the discussion that the livers of adults ranged in weight from 0.01-0.15g, so error due solely to the balance ranged from 0.01/0.01*100 = 100% to 0.01/0.15 = 7%. Thus, the error in the liver weights may explain the relatively poor relationship in Fig. 1d, rather than any actual issue with HSI as a predictor of lipid content. In my opinion the HSI data are too inaccurate to be published and should be excluded from the paper. Second, both the field and laboratory methods are inadequately described (please see specific comments for details). Third, there was substantial redundant information repeated throughout the paper. This needs to be substantially reduced (e.g., the table descriptions contain several details that are in the methods). Fourth, one of the conclusions made by the authors, that hepatosomatic index is not a good predictor of lipid content in juvenile Slimy Sculpin, is absolutely unsupported by the study. Hepatosomatic index of juveniles was not obtained in the study because the balance was too inaccurate, so how can the validity of HSI for juveniles be assessed? Finally, the writing needs to be improved in places that I have detailed below.
Specific comments
Lines 19-20 Overly specific sentence. I suggest “And while Fulton's K did not perform quite as well, we consider it a sufficient substitute if the data necessary to calculate Kn are unavailable.”
Line 77 This sentence is unclear. I suggest: “We expected, therefore, to see a stronger relationship between Ks than Fulton’s K with lipid content.”
Line 90 “presumably to diminish susceptibility to size-selective predators” seems like quite a leap to me. It assumes that length helps more than body mass to discourage predators. I’d delete it or justify.
Line 95 “…New Brunswick, Canada.”
Lines 102-105 Please split this sentence into two.
Line 113 To be 100% clear please specify which measurements. Weight and length?
Line 110 The methods described in this paragraph are completely unclear to me. It says that 3 fish were selected from each of the length categories for measurement. There are 4 length bins, which adds up to 12 fish, but it says lower in the paragraph that 96 fish were processed. Also, it seems like 2 weights were taken, one set with the fish alive and one with the fish dead, but I can’t be sure based on this paragraph. On line 113 what is meant by ‘re-measured’? Were the fish measured already? This needs to be explained. How were the fish weighed the first time? I assume on a balance in a cup of water since they were alive? This paragraph needs far more detail. I suggest having one paragraph describe the methods for lethally processing the fish, and a second paragraph for processing the fish non-lethally. Were the non-lethally processed fish returned to the streams?
Line 139 I believe that accuracy here should be sensitivity
Lines 141-151 The description of these methods seems inadequate. What did the centrifugation accomplish? How were the lipids separated from the total volume after centrifugation?
Line 147 ‘run’ of what? Lipids? What is a sample (I assume the 200 mg fish homogenates?)
Line 154: ‘online’ maybe unnecessary?
Line 158: Why not include ‘non-lethal’ n here rather than referring to another table?
Lines 156-163 This seems like too much information for the table description. A lot of it is redundant with the methods.
Lines 174 There is redundancy here. No need to mention dates and sample locations again.
Lines 183-185 We know where the samples were collected and we know the dates!
Line 202 ‘proportion’ would be a better word than ‘fraction’
Line 220 Missing period?
Line 217 What was the response variable for the five models? Lipid content?
Line 228 Delete ‘are’?
Line 242 Why ‘seemingly more so’? Is the slope higher? Correlation tighter?
Line 365 This seems unnecessary: “meaning that with an increase in condition there was a corresponding increase in 366 percent lipid” Readers will know what a positive correlation means.
Line 391 Is ± a SE? 95% CI? SD? Please specify.
Line 393 ‘Requirements’ seems like a poor word choice since it implies that a fish without the required amount of fat cannot persist in lentic or lotic environments.
Line 419 What is meant by ‘superficial water’?
Line 424 Please replace ‘This’ with something more specific (i.e., The similarity in predictive utility of lipids among the morphometric indices…)
Line 430 The writing here and below is pretty marginal. I suggest choosing a stance and arguing it. Like, argue that storing the samples at -80C had no influence on triglycerides, don’t first say it is possible that storage at -80C affected them and then say it probably didn’t.
Line 452 What did you not see a difference in between males and females? Please clarify.
Line 459 Concluding that for C. cognatus, hepatosomatic index “Is of little use when studying juveniles” is an outrageous conclusion to draw from this study. The scale was not sensitive enough to weigh the livers of the juveniles, so the utility of hepatosomatic index as a predictor of lipid content could not be tested for juveniles. Even the test in adults seemed marginal at best. I would conclude that you need a more sensitive balance.
Line 470 Better to just say “in the case of Ks it too requires…”
Line 472 Like in the abstract, why ‘historical data’? Aren’t all data historical once collected?
Line 483-87 Sentence too long
Author Response
Response to Reviewer 3 Comments
Point 1: First, I am concerned with the validity of the HSI analysis because the scale used in the field was so insensitive (+- 0.01 g). The authors report in the discussion that the livers of adults ranged in weight from 0.01-0.15g, so error due solely to the balance ranged from 0.01/0.01*100 = 100% to 0.01/0.15 = 7%. Thus, the error in the liver weights may explain the relatively poor relationship in Fig. 1d, rather than any actual issue with HSI as a predictor of lipid content. In my opinion the HSI data are too inaccurate to be published and should be excluded from the paper.
Response 1: We acknowledge that the balance used was not well suited to weighing the livers of a fish this size. As a result, we will remove all mention of HSI from this manuscript. Given that morphometric indices are more commonly used we do not think the removal of HSI substantially detracts from the value of this manuscript.
Point 2: Second, both the field and laboratory methods are inadequately described (please see specific comments for details).
Response 2: Please see our response to each below.
Point 3: Third, there was substantial redundant information repeated throughout the paper. This needs to be substantially reduced (e.g., the table descriptions contain several details that are in the methods).
Response 3: We have removed a substantial amount of information from each table and figure description. If there remains any information you would like us to remove, please let us know. We have left descriptions for any abbreviations used so that each figure and table may be understood without reference to the article.
Point 4: Fourth, one of the conclusions made by the authors, that hepatosomatic index is not a good predictor of lipid content in juvenile Slimy Sculpin, is absolutely unsupported by the study. Hepatosomatic index of juveniles was not obtained in the study because the balance was too inaccurate, so how can the validity of HSI for juveniles be assessed? Finally, the writing needs to be improved in places that I have detailed below.
Response 4: As above, we have removed all mention of HSI.
Point 5: Lines 19-20 Overly specific sentence. I suggest “And while Fulton's K did not perform quite as well, we consider it a sufficient substitute if the data necessary to calculate Kn are unavailable.”
Response 5: Lines 18-19. We have edited the lines in question to match what was suggested.
Point 6: Line 77 This sentence is unclear. I suggest: “We expected, therefore, to see a stronger relationship between Ks than Fulton’s K with lipid content.”
Response 6: Lines 72-73. We have edited the lines in question for additional clarity.
Point 7: Line 90 “presumably to diminish susceptibility to size-selective predators” seems like quite a leap to me. It assumes that length helps more than body mass to discourage predators. I’d delete it or justify.
Response 7: Deleted.
Point 8: Line 95 “…New Brunswick, Canada.”
Response 8: Added
Point 9: Lines 102-105 Please split this sentence into two.
Response 9: Now split in two.
Point 10: Line 113 To be 100% clear please specify which measurements. Weight and length?
Response 10: We added “length and mass” in front of “measurements” to clarify.
Point 11: Line 110 The methods described in this paragraph are completely unclear to me. It says that 3 fish were selected from each of the length categories for measurement. There are 4 length bins, which adds up to 12 fish, but it says lower in the paragraph that 96 fish were processed. Also, it seems like 2 weights were taken, one set with the fish alive and one with the fish dead, but I can’t be sure based on this paragraph. On line 113 what is meant by ‘re-measured’? Were the fish measured already? This needs to be explained. How were the fish weighed the first time? I assume on a balance in a cup of water since they were alive? This paragraph needs far more detail. I suggest having one paragraph describe the methods for lethally processing the fish, and a second paragraph for processing the fish non-lethally. Were the non-lethally processed fish returned to the streams?
Response 11: Lines 89-124. Substantially altered to better explain methods for how fish were caught. Paragraphs for how fish were lethally and non-lethally processed are now separate, as suggested.
Point 12: Line 139 I believe that accuracy here should be sensitivity
Response 12: We meant to use readability (we have replaced “accuracy” with what we measured to (i.e. nearest 0.01 g) throughout the manuscript).
Point 13: Lines 141-151 The description of these methods seems inadequate. What did the centrifugation accomplish? How were the lipids separated from the total volume after centrifugation?
Response 13: Line 136-146. Further details regarding lipid extraction, which include answers to the above questions, have been included.
Point 14: Line 147 ‘run’ of what? Lipids? What is a sample (I assume the 200 mg fish homogenates?)
Response 14: We have removed the use of word run.
Point 15: Line 154: ‘online’ maybe unnecessary?
Response 15: Sentence removed.
Point 16: Line 158: Why not include ‘non-lethal’ n here rather than referring to another table?
Response 16: n now included in Table 1.
Point 17: Lines 156-163 This seems like too much information for the table description. A lot of it is redundant with the methods.
Response 17: All Tables now include less information.
Point 18: Lines 174 There is redundancy here. No need to mention dates and sample locations again.
Response 18: Removed mention of dates and sample locations from Table 1
Point 19: Lines 183-185 We know where the samples were collected and we know the dates!
Response 19: Removed mention of dates and sample locations from Table 2
Point 20: Line 202 ‘proportion’ would be a better word than ‘fraction’
Response 20: Line 188: Replaced ‘fraction’ with ‘proportion’
Point 21: Line 220 Missing period?
Response 21: Didn’t catch what was being referred to, but we have modified the sentence slightly. Please let us know if it requires further editing.
Point 22: Line 217 What was the response variable for the five models? Lipid content?
Response 22: Line 204. Yes, we have added this information for clarity.
Point 23: Line 228 Delete ‘are’?
Response 23: Leaving as ‘were’ throughout to keep in past tense.
Point 24: Line 242 Why ‘seemingly more so’? Is the slope higher? Correlation tighter?
Response 24: Deleted to avoid confusion.
Point 25: Line 365 This seems unnecessary: “meaning that with an increase in condition there was a corresponding increase in 366 percent lipid” Readers will know what a positive correlation means.
Response 25: Sentenced referred to now deleted.
Point 26: Line 391 Is ± a SE? 95% CI? SD? Please specify.
Response 26: Lines 369-374. Sentence removed.
Point 27: Line 393 ‘Requirements’ seems like a poor word choice since it implies that a fish without the required amount of fat cannot persist in lentic or lotic environments.
Response 27: Sentence removed.
Point 28: Line 419 What is meant by ‘superficial water’?
Response 28: This was in reference to the water that ‘clings’ (for want of a better word) to the surface of fish after you take them out of water and then weigh them. In retrospect, this comment was highly speculative, and as a result, removed.
Point 29: Line 424 Please replace ‘This’ with something more specific (i.e., The similarity in predictive utility of lipids among the morphometric indices…)
Response 29: Line 366. Written as suggested.
Point 30: Line 430 The writing here and below is pretty marginal. I suggest choosing a stance and arguing it. Like, argue that storing the samples at -80C had no influence on triglycerides, don’t first say it is possible that storage at -80C affected them and then say it probably didn’t.
Response 30: In retrospect we realise this section was irrelevant and therefore removed
Point 31: Line 452 What did you not see a difference in between males and females? Please clarify.
Response 31: This sentence was out of place and therefore removed.
Point 32: Line 459 Concluding that for C. cognatus, hepatosomatic index “Is of little use when studying juveniles” is an outrageous conclusion to draw from this study. The scale was not sensitive enough to weigh the livers of the juveniles, so the utility of hepatosomatic index as a predictor of lipid content could not be tested for juveniles. Even the test in adults seemed marginal at best. I would conclude that you need a more sensitive balance.
Response 32: Agreed. Please see ‘response 1’
Point 33: Line 470 Better to just say “in the case of Ks it too requires…”
Response 33: Line 385. Reworded.
Point 34: Line 472 Like in the abstract, why ‘historical data’? Aren’t all data historical once collected?
Response 34: A fair point. We removed use of the term from both the abstract and conclusion, though we have kept it on lines 76 and 102.
Point 35: Line 483-87 Sentence too long
Response 35: Line 397-401. Shortened.
Thank you for your comments.

Round 2
Reviewer 3 Report
The manuscript is much improved overall and I think it is suitable for publication. I have some relatively minor suggestions below, mostly just meant to clarify certain sentences.
Line 44 I would change ‘are’ to ‘can be’. Also, another disadvantage of whole body lipid extraction is that other biomarkers cannot be measured once the entire fish is extracted for lipids.
Line 54 Delete ‘so as’
Line 102 Missing a ‘the’ after ‘be’
Line 129 I believe ‘were’ should be ‘was’ after C. cognatus
Line 54-55 and lines 143-144 In the intro it says that somatic mass doesn’t include stomach contents or gonad, while in the methods it says that somatic mass was determined by removing the gonad and measuring mass (no mention of stomach content removal). Please reconcile these statements.
Line 110 Please clarify somewhere in Materials and Methods whether ‘whole fish lipid extraction’ measurements included the excised gonad or not.
Line 193 It would make the methods easier to follow if it said something like ’A subsample of 10 of the 68 adults were analyzed in triplicate’. Please insert what the purpose was. I assume it was to quantify within individual variability?
Line 197 What kind of contaminants? Lipid?
Line 568 Awkwardly written: ‘though each index accounted for slightly more variation’
Line 569 Lower than what?
Line 630 I suggest inserting ‘only’ between ‘displayed’ and ‘slight’. Also replace ‘should be’ with ‘is’
Line 635 I’m still confused to the applicability of stomach contents. They were ignored in this study, correct? It’s a bit disconcerting to have stomach contents come up in the intro and discussion but not in the methods and results.
Line 684 Suggest deleting ‘in question’
Line 699-700 I’d delete the last sentence of this paragraph: ‘whether they should or not…’. Seems unnecessary.
Line 778-779 So the results are only applicable to spawning season, since all other seasons are ‘prior to spawning’? That doesn’t seem correct. Maybe ‘immediately prior’ or something would be better?
Author Response
Point 1: Line 44 I would change ‘are’ to ‘can be’. Also, another disadvantage of whole body lipid extraction is that other biomarkers cannot be measured once the entire fish is extracted for lipids.
Response 1: Changed to 'can be'. While we agree with the additional point made, we feel it would be out of place with the sentence in question (which is about direct methods in general rather than just total lipid extraction).
Point 2: Line 54 Delete ‘so as’
Response 2: Deleted 'so as' from lines 50 and 111.
Point 3: Line 102 Missing a ‘the’ after ‘be’
Response 3: Added 'the'.
Point 4: Line 129 I believe ‘were’ should be ‘was’ after C. cognatus
Response 4: While there is a convincing argument to regard scientific names as singular, they can be (and often are) regarded as either singular or plural. Further, we feel 'were' reads better than 'was'.
Point 5: Line 54-55 and lines 143-144 In the intro it says that somatic mass doesn’t include stomach contents or gonad, while in the methods it says that somatic mass was determined by removing the gonad and measuring mass (no mention of stomach content removal). Please reconcile these statements.
Response 5: Apologies for the confusion. We removed both the gonads and stomach contents. Lines 114 to 115 now read 'After the stomach contents and gonads were excised, each carcass was weighed to determine its somatic mass. '
Point 6: Line 110 Please clarify somewhere in Materials and Methods whether ‘whole fish lipid extraction’ measurements included the excised gonad or not.
Response 6: On line 136 (following 'whole fish carcasses') we have added 'including stomach contents and gonads'
Point 7: Line 193 It would make the methods easier to follow if it said something like ’A subsample of 10 of the 68 adults were analyzed in triplicate’. Please insert what the purpose was. I assume it was to quantify within individual variability?
Response 7: Yes. Lines 149 - 151 now read 'A subsample of 10 adults were analysed in triplicate to quantify within-individual variability. Values were later averaged. Owing to the amount of homogenate required, triplicates came from individuals >70 mm in total length (34 of the 68 adults). '
Point 8: Line 197 What kind of contaminants? Lipid?
Response 8: Yes. Line 154 changed to 'presence of contaminants (i.e. lipid).'
Point 9: Line 568 Awkwardly written: ‘though each index accounted for slightly more variation’
Response 9: Agreed. We have split the sentences in two. Lines 283 - 285 now read 'Juvenile percent lipid values were also positively correlated with all three indices (Figure 3). Interestingly, each index accounted for slightly more variation than their adult counterparts'
Point 10: Line 569 Lower than what?
Response 10: Lines 285-286 changed to 'The RMSE values associated with the juvenile data were also lower than those of the adults'
Point 11: Line 630 I suggest inserting ‘only’ between ‘displayed’ and ‘slight’. Also replace ‘should be’ with ‘is’
Response 11: Lines 331-333 now read 'And given that C. cognatus displayed only slight allometry, it is unsurprising that Kn performed only marginally better.'
Point 12: Line 635 I’m still confused to the applicability of stomach contents. They were ignored in this study, correct? It’s a bit disconcerting to have stomach contents come up in the intro and discussion but not in the methods and results.
Response 12: We have updated: lines 71, 80 and 114 in the methods section; and the descriptions for Tables 1, 3, and 4, and Figures 1, 2, 3 and 4 in the results section to acknowledge that somatic mass includes neither the stomach contents nor the gonads.
Point 13: Line 684 Suggest deleting ‘in question’
Response 13: Line 355. Deleted 'in question'. Now reads 'Conversely, studies with values of ≤ 0.25 do not seem to result in an index being recommended'
Point 14: Line 699-700 I’d delete the last sentence of this paragraph: ‘whether they should or not…’. Seems unnecessary.
Response 14: Line 362. Agreed. Deleted the following 'Whether they should or not is another matter entirely and beyond the scope of this article.'
Point 15: Line 778-779 So the results are only applicable to spawning season, since all other seasons are ‘prior to spawning’? That doesn’t seem correct. Maybe ‘immediately prior’ or something would be better?
Response 15: Agreed. Lines 393 to 395 changed to 'We would not, for example, recommend the use of such indices immediately prior to spawning (which, in New Brunswick, occurs in mid to late May). '
Thank you again for your constructive comments.